# Development and Evaluation of a Blocking Lateral Flow Assay Strip for Detection of Newcastle Disease Virus Antibodies

**DOI:** 10.3390/vetsci10020152

**Published:** 2023-02-13

**Authors:** Rongzhou Lv, Junqing Guo, Yuhang Zhang, Xun Wang, Ge Li, Zekun Meng, Li Wang, Shujun Chai, Qingmei Li, Gaiping Zhang

**Affiliations:** 1College of Veterinary Medicine, Henan Agricultural University, Zhengzhou 455046, China; 2Key Laboratory of Animal Immunology, Henan Academy of Agricultural Sciences, Zhengzhou 450002, China; 3Jiangsu Co-Innovation Center for Prevention and Control of Important Animal Infectious Disease and Zoonose, Yangzhou University, Yangzhou 225009, China

**Keywords:** Newcastle disease virus, blocking lateral flow bLFA strip, neutralizing antibodies, HI, monoclonal antibodies

## Abstract

**Simple Summary:**

Newcastle disease (ND) is an Office International Des Epizooties (OIE) notifiable terrestrial and aquatic animal disease that can infect chickens of all ages with a high mortality rate, causing ongoing damage to the poultry industry in China currently. In this study, we have developed a blocking lateral flow assay (bLFA) strip for the rapid detection of Newcastle disease virus (NDV) antibodies, using neutralizing monoclonal antibodies against hemagglutinin-neuraminidase. The results showed that the chicken NDV hyperimmunized serum had a complete blocking titer of 11 log 2, and half-blocking titer of 13 log 2, which are 4 times less than and the same as that of the Hemagglutination inhibition (HI) test (13 log 2), and 8 and 2 times less than that of the virus neutralization test (14 log 2), respectively. Additionally, the bLFA strip has no cross-reactivity with positive serum of other avian pathogens. A total of 510 clinical samples were tested for NDV antibodies. The coincidence rate between the results of the bLFA strip and HI test was 97.65%. Therefore, it is an ideal alternative method for assessing the clinical immunity of ND vaccines in the field in real-time.

**Abstract:**

Newcastle disease (ND) is an acute septicemic infectious disease caused by Newcastle disease virus (NDV). Considering that vaccination is currently the main modality for the prevention of ND, it is essential to assess the effectiveness of clinical immunization. In this study, we have developed a blocking lateral flow assay (bLFA) strip for the rapid detection of NDV antibodies using the monoclonal antibody 9C1 against haemagglutinin-neuraminidase (HN), which allows for the determination of an NDV-specific antibody titer within 10 min at room temperature. In addition, the bLFA strip has no cross-reactivity with the positive serum of other avian pathogens including avian influenza subtypes H5, H7, and H9, MD, IBD, IB, EDS, and avian adenovirus. The ability of the bLFA strip for detecting a neutralizing antibody was also estimated. The results showed that the chicken NDV hyperimmunized serum had a complete blocking (100%) titer of 11 log 2, and half-blocking titer of 13 log 2, which are 4 times less than and the same as that of the HI test (13 log 2), and 8 and 2 times less than that of the VN test (14 log 2), respectively. A total of 510 clinical samples were tested for NDV antibodies. The coincidence rate between the results of the bLFA strip and HI test was 97.65%. Therefore, it is an ideal alternative method for assessing the clinical immunity of ND vaccines in the field in real-time.

## 1. Introduction

Newcastle disease virus (NDV) is a membrane-bearing, non-segmented, negative single-stranded RNA virus, belonging to the genus *Paramyxoviridae* [1]. Its genomic size is 15 kb, encoding six structural proteins (NP, P, M, F, HN, and L), and two non-structural proteins (V and W) [2]. F and HN are the two major vesicular membrane glycoproteins on the surface of NDV, playing important roles in viral infection [3,4]. As a notifiable epidemic disease listed by the World Organization for Animal Health (OIE), NDV can infect chickens of all ages with a high mortality rate, still causing ongoing damage to the poultry industry in China currently. It mainly causes respiratory distress, neurological disorders, hemorrhage, and necrosis of mucous and serous membranes in birds [5,6,7].

At present, the prevention and control of ND in China is still dominated by vaccination [8,9]. Maternal antibodies in chicks are vital to influencing the effectiveness of ND vaccines (especially attenuated vaccines). The monitoring of maternal antibody and immune antibody levels in chickens is an important basis for developing and optimizing NDV immunization programs [10,11,12]. Meanwhile, the serum antibody levels in vaccinated chickens, especially neutralizing antibody levels, are important indicators for evaluating NDV vaccination [13,14,15]. The Hemagglutination inhibition (HI) test remains a common method for detecting NDV antibodies and can effectively evaluate the level of immune antibodies and immune protection against the vaccine [16,17,18,19,20]. However, the HI test is complex and time-consuming to perform and requires specific specialist operators. Due to the complicated process, it is usually limited to the laboratory and cannot be popularized and applied in field detection. The rapid detection technology of the lateral flow assay (LFA) is specific, sensitive, simple, rapid, and low-cost, which does not depend on laboratory conditions to achieve a “foolproof” operation [21,22,23,24]. Studies have shown that using NDV (LaSota strain) purified virus-labeled colloidal gold, the strip can detect the lowest HI antibody titer of 4 log 2, and the coincidence rate with HI is 95.9% [25]. When colloidal gold was labeled with murine anti-chicken IgG Fc mAb, the strip was 1 titer less sensitive than HI with a 96.9–99.1% overlap [26].

We used NDV-neutralizing mAb to establish the immunochromatographic test blocking mode and developed a blocking lateral flow assay (bLFA) to detect the level of NDV-neutralizing antibody, which can be used to evaluate NDV maternal antibody and the immunization effect of vaccination.

## 2. Materials and Methods

### 2.1. Viruses and Antibodies

The NDV standard strain F_48_E_8_ was obtained from the China Institute of Veterinary Drug Control. NDV LaSota vaccine strains, chicken NDV hyperimmunized serum, and monoclonal antibodies (mAbs) against the HN protein were identified previously [27]. Specific pathogen-free (SPF) chicken and embryos were purchased from Beijing Meriavitong Experimental Technology Co., Ltd., Beijing, China, and were incubated in incubators for 9–11 days for NDV proliferation. Standard positive sera for H5, H7, and H9 subtypes of avian influenza (AI), chicken egg drop syndrome (EDS), infectious bronchitis (IB), and Newcastle disease (ND) were purchased from Harbin Guosheng Biotechnology Co., Ltd., Harbin, China. Positive sera for chicken Marek’s disease (MD), chicken infectious bursal disease (IBD), and avian adenovirus (AD) were stored in a laboratory. The LaSota vaccine was purchased from Guangxi Liyuan Biology, Nanning, China. The mAbs are screened and preserved in the laboratory. Additionally, mAbs ascites are purified using caprylic acid and ammonium sulfate precipitation [28] and the concentrations were determined using a spectrophotometer.

### 2.2. Preparation of Antigen for Detection

The expressed HN protein of transgenic rice was purified by nickel affinity chromatography, ion exchange chromatography, and gel filtration chromatography to obtain the HN-expressed protein with a 95% purity, and the recombinant antigen of the HN protein was prepared [29].

The NDV LaSota strain was inoculated into the allantoic cavity of SPF chicken embryos at 9–11 days of age. The allantoic fluid was collected at 36–120 h and centrifuged at a low speed of 3000 r/min for 30 min at 4 °C. The supernatant was collected, filtered through 0.45 and 0.22 μm filters, then packed and stored at −80 °C.

The NDV allantoic fluid (300 μL) was inactivated at 4 °C for 24 h, followed by 37 °C for 1 h, and in a water bath at 56 °C for 1 h, respectively. The inactivated allantoic fluid was taken to determine whether the virus was completely inactivated. The HA titer of inactivated virus was determined by an HA test.

### 2.3. Monoclonal Antibody Pairing Test and Principles of bLFA Strip

Ten mAbs labeled with colloidal gold were paired with chicken anti-NDV polyantibody IgG, 1 μL of labeled colloidal gold was placed on the conjugate pad, and the concentration of chicken anti-NDV polyantibody IgG was diluted to 1 mg/mL for T-line blot, and 0.3 mg/mL staphylococcal protein A (SPA) was blotted for C-line. The NDV strain LaSota (HA = 6 log 2) was diluted twice in a dilution cup with 100 μL per well and a negative control cup with PBS only was set up. The labeled mAbs were screened by detecting the color intensity of the T-line and comparing it with the HA test, respectively.

Six mAbs that could pair with chicken anti-NDV polyantibody IgG were selected, and the mAb with the best blocking effect was selected according to the blocking mode and compared with HI. A total of 1 μL of labeled colloidal gold was dotted on the conjugate pad, and the film blotted with quality control C-line and detection T-line was formed as a bLFA strip. The chicken NDV hyperimmunized serum (HI = 13 log 2) was 2-fold serially diluted in PBS, 50 μL per well, and then the LaSota strain (HA = 6 log 2) was added to the diluted hyperimmunized serum according to the color strength in the previous step, 50 μL per well, and the bLFA strip was inserted to determine the reading value of the bLFA strip. The well containing LaSota antigen without a serum sample was set as the blank control. The blocking efficiency was determined by comparing T-line color development with the blank control and compared with hyperimmunized serum HI. The area of the absorbance value was calculated by the bLFA strip reader to calculate the blocking rate (PI (%) = [(ROD_Blank_ − ROD_Sample_)/ROD_Blank_ × 100%], and the mAb with the best blocking effect was selected for the subsequent tests (Figure 1).

The bLFA strip detects NDV antibodies using a competitive later flow assay based on the neutralizing mAb against the HN protein. The serum sample mixed with the detection antigen dissolves the gold-labeled mAb in the conjugate pad and moves through the detection membrane from the sample end to the absorbent ones by capillary action after the solution is applied to the bLFA strip. During the movement of the mixture solution, the detection antigen is recognized by the gold-labeled mAb and captured by the chicken anti-NDV polyantibody IgG immobilized on the membrane, forming a red T-line, and the excess gold-labeled mAbs are trapped by the SPA blot, forming a red C-line, which forms two red-colored lines on the detection zone, showing a negative result of the NDV antibodies. If an NDV-specific antibody is present in the sample, it would compete or block the binding of gold-labeled mAb and a detection antigen, resulting in the weaker or complete disappearance of the T-line on the membrane, indicating a positive result of NDV-neutralizing antibodies.

### 2.4. Optimization of Antibody Blocking Detection Method

The selection and combination of influencing factors in the blocking bLFA strips were tested separately. At first, 10, 50, and 100 µg of the ascitic fluid of mAbs were added into 1 mL of colloidal gold solution to determine the conjugation ratio of mAb with colloidal gold. Then, gold-labeled mAb was micro-sprayed onto a conjugate pad at 3.75, 5, and 7.5 µL/cm at a position to determine the working concentration of gold-labeled mAb. Subsequently, the working concentration of anti-NDV polyantibody IgG (T-line) was determined by micro-spraying onto a nitrocellulose membrane at 3.75, 5, and 7.5 µL/cm on the bLFA strip. The strips were then assembled at the optimal working concentration and used for optimizing the detection antigen. The chicken NDV hyperimmunized serum was tested using a live and inactivated NDV LaSota strain and the recombinant HN protein at different concentrations as the detection antigens, respectively. Finally, the reaction time for the detection antigen and the serum sample was determined by incubating the NDV hyperimmunized serum with the detection antigen at room temperature for 3, 5, and 10 min compared with the HI test.

### 2.5. Sensitivity of the bLFA Strip

In order to evaluate the sensitivity of the bLFA strip, the NDV hyperimmunized serum was 2-fold serially diluted from 1 log 2 to 14 log 2 and tested by the bLFA strip compared with HI and virus neutralization tests.2.6. HA and HI Test

In the HI experiment, the HA titer of the NDV LaSota strain was measured according to the standard HA experiment procedure [30], and the HI experiment was carried out using the 4 HA unit (HAU) virus. The serum was 2-fold serially diluted to 25 μL per well, and 25 μL of 4 HAU virus diluent was added to each well, then the reaction was performed at room temperature for 10 min. Then, 25 μL 1% red blood cell suspension was added to each well, and the reaction time was 30 min at room temperature. The serum HI titer was determined by erythrocyte agglutination.

### 2.6. Virus Neutralization Test

The micropore virus neutralization test was performed using BHK21 cultured in 96-well cell plates. Briefly, the 50% tissue culture infective dose (TCID_50_) of the viral solution was first measured, 100 TCID_50_ of virus was incubated with a 2-fold serially diluted serum sample in a 96-well plate at 37 °C for 1 h, then the mixtures were added to the cells and incubated at 37 °C for two days. The wells were fixed with pre-chilled anhydrous ethanol and blocked with 5% skimmed milk at 37 °C for 2 h. The anti-HN mAb (1:1000 diluted) and the goat anti-mouse IgG/HRP secondary antibody (1:5000 diluted) were subsequently added into the wells and incubated for 1 h at 37 °C to detect the NDV-infected cells. The cells were washed three times with PBS containing 0.05% Tween-20 between each step. The AEC substrate solution was then added for color development and observed under a microscope. The highest dilution of the serum that completely blocked the NDV infection in the cells is considered as its neutralizing antibody titer.

### 2.7. Specificity of the bLFA Strip

The standard positive sera of H5, H7, and H9 subtypes of AI, MD, IBD, IB, EDS, AD, ND, and chicken negative were diluted at 1:2 in PBS and tested by the bLFA strips. Meanwhile, the well containing a detection antigen without a serum sample was set as the blank control.

### 2.8. The Dynamic Change of Anti-NDV Antibody in Infected and Vaccinated Chickens

In total, 5 15-day-old SPF chickens were experimentally challenged with 100 TCID_50_ of the NDV standard strain F_48_E_8_ by the nasal route, and five chickens were immunized with the LaSota vaccine according to the manufacture’s instruction. Serum samples were collected from the F_48_E_8_-infected and LaSota-vaccinated chickens when they were 1, 2, 4, 5, 9, and 13 days post-infected (DPI) and they were tested by the bLFA strips and HI test, respectively.

### 2.9. The bLFA Strip Stability

To determine the storage life, the bLFA strips were placed in a plastic bag sealed with desiccant. The bLFA strips were then examined for specificity and sensitivity by testing the NDV hyperimmunized serum as well as AI H7 positive sera and negative sera at 0, 3, 6, 9, 12, and 18 months after storage.

### 2.10. Clinical Application of Newcastle Disease Antibody bLFA Strip

A total of 510 clinical samples (including 434 sera and 76 yolks) from different chicken farms in Henan province, China, were examined by the bLFA strip and HI test. Since the serum with an HI titer above 4 log 2 is often judged as the serum with protective potency in clinical practice, 510 clinical samples were diluted 1:4. If it can be completely blocked, the T-line is judged as positive without color development, and the HI titer is compared to judge the coincidence rate of the bLFA strip.

## 3. Results

### 3.1. Establishment of the bLFA Strip

The bLFA strip for detecting NDV-neutralizing antibody was established using the anti-HN mAbs and the detection antigen after optimizing the working conditions. The mAb 9C1 with NDV-neutralizing activity was selected for the gold-labeled mAb, with which the bLFA strip detected the highest titers of LaSota strain (7 log 2) and NDV antibody (12 log 2) (Table 1). As shown in Table 2, the optimized working conditions of the bLFA strips were further determined by optimizing the various influencing factors for NDV antibody detection.

### 3.2. Procedures of bLFA Strip

The detection steps of the bLFA strip are as follows: 50 µL of the serum sample (1:4 diluted) is firstly incubated with 50 µL of the dilution solution containing HN protein (0.14 ng/µL) for 10 min; the mixture is then added to the sample well of the bLFA strip and placed horizontally at room temperature for 10 min to observe the result. If both the T-line and C-line appear on the membrane as the same as the blank control, the sample was recorded as negative, indicating the absence of NDV antibodies. When the T-line completely disappeared or weakened significantly compared with the blank control, it is considered a positive result, indicating the presence of the NDV-neutralizing antibodies in the sample.

### 3.3. Sensitivity of the bLFA Strip

The two-fold serially diluted chicken anti-NDV hyperimmunized serum samples were tested by the bLFA strips, and the ROD data of the colored membranes were obtained by TSR-3000 reader screening (Bio-Dot, Irvine, CA, USA). Meanwhile, the serum antibody titers were parallelly determined by HI and VN tests. The results showed that the NDV antibody titer detected by the bLFA strips was determined as 11 log 2 with complete blocking (100%) and 13 log 2 with half-blocking (50%) (Figure 2), which are 4 times less than and the same as that of the HI test (13 log 2), and 8 and 2 times less than that of the VN test (14 log 2), respectively (Table 3).

### 3.4. Specificity of the bLFA Strip

In order to verify the specificity of the bLFA strip, Avian influenza subtypes H5, H7, and H9, IB, EDS, MD, AD, ND positive sera, and negative serum were tested by the bLFA strips. The results showed that the ND serum entirely blocked the color of the T-line, whereas the T-line color of the other sera of the related avian diseases showed no significant difference compared with the blank control, indicating that the blocking test was highly specific for an ND antibody with no cross-reaction with that of the other avian diseases (Figure 3).

### 3.5. The Dynamic Change of Anti-NDV Antibody in Infected and Vaccinated Chickens

The F_48_E_8_-infected chickens showed obvious nervous signs of torticollis and opisthotonus at 3 DPI, there was the occurrence of diarrhea and respiratory distress at 4 DPI, and they began to die at 5 DPI, whereas the LaSota-vaccinated chickens were healthy without any clinical signs. We collected the serum samples of 0, 1, 2, 4, 5, 9, and 13 DPI, which were tested by the bLFA strip compared with the HI test. In the LaSota-vaccinated chickens, the NDV-specific antibodies were detected at 5 DPI by both bLFA strip and HI tests and they reached a high level at 13 DPI (Figure 4). In the F_48_E_8_-infected chickens, the blocking test bLFA strips detected NDV-specific antibodies at 4 DPI, which is later than that of the HI test (2 DPI). Although the antibody titers of the bLFA strip in the infected and vaccinated chickens were slightly lower that of HI, the overall trend of the NDV antibody is similar, indicating that the bLFA strip could be an alternative simple method for the evaluation of NDV vaccination.

### 3.6. The Storage Life

Stored at 4 °C for 18 months, the bLFA strips were sampled to determine their specificity and sensitivity as well as their appearance. The results show that the bLFA strips sealed in plastic bags with a desiccant can be stored for at least 12 months at 4 °C without any loss of sensitivity (Table 4).

### 3.7. Detection of Newcastle Disease Antibody in Clinical Specimens

To validate the bLFA strips for clinical application, 434 serum and 76 yolk samples diluted at 1:4 were tested by the bLFA strips and HI tests. Ninety-five percent (485/510) of the tested samples were shown to be positive by the bLFA strips, indicating that the vaccination had a positive effect in the field. Using the HI titer (4 log 2) as a reference, the diagnostic sensitivity (DSn), diagnostic specificity (DSp), and accuracy of the bLFA strip were calculated as 96.04%, 100%, and 97.65% according to the formula: DSn = TP/(TP + FN); DSp = TN/(TN + FP), and the accuracy = (TP + TN)/total number of samples tested, in which TP, FP, TN, and FN indicated true positive, false positive, true negative, and false negative, respectively (Table 5).

## 4. Discussion

The blocking lateral flow assay (bLFA) is similar in principle to the blocking ELISA (bELISA). The conventional bLFA strip is an indirect method to detect antibodies, which can only detect the total antibody level rather than the neutralizing antibody. We developed a blocking mode based on a broad-spectrum neutralizing mAb blocking test, and it achieves a 97.65% compliance with the standard test for antibodies, HI, and can be used instead of the categorical HI to evaluate maternal antibody and immune protection from vaccines. Although there is a slight delay in the detection of antibodies by bLFA compared to HI, the overall trend is the same and bLFA can detect the appearance of antibodies at the onset of disease in chickens, allowing for an early warning. The short time required for bLFA makes it easier to use in field testing. The bLFA strips we have developed can therefore be used for preliminary screening in the field to reduce the sample volume for more sophisticated tests, such as VN tests and RT-PCR in the laboratory, thus enabling a faster monitoring of outbreaks and vaccine immunity levels. There is no covalent bond formation between gold particles and antibody molecules in gold-labeled antibodies. The two are combined by a van der Waals force between opposite charges. Colloidal gold has little effect on the reactivity of labeled antibodies and has a high labeling rate. The test results can be determined by adding the sample to the bLFA strip and adding it to the sample well in 5–10 min. However, the routine HI and ELISA tests are not only complicated to operate, but also require 1–2 h of detection time, so bLFA stripsare is significantly better than HI and ELISA tests. Additionally, the determination of the result is simple; all that needs to be compared is whether the T-line appears as a red stripe and its strength can be intuitive and accurate, and the image of the judge is simple and clear with a false negative and false positive miscalculation difficult to obtain.

As one of the common diseases affecting the poultry industry in China, Newcastle disease is difficult to completely kill, the inactivity of the viruses in the contaminated environment is difficult to obtain, and chickens can still excrete the viruses into the environment after a chronic passage, which is harmful to healthy chickens. Even individuals with HI titers greater than 6 log 2 can still be coinfected with virulent strains, and the reason may be related to the inconsistency between HI antibody levels and ND protection. HI is merely an easy detection of the antibody concentration index in measuring the chicken resistance strength level, but it only detects the circulating antibodies (IgG) level; the antibody levels (IgA) in the respiratory tract were not detected [31]. The VN test is the most sensitive and specific neutralizing antibody detection method, and the level of the NDV-neutralizing antibody is directly related to immune protection. However, this technology method is not only time-consuming, cumbersome, and expensive, but also must be operated in a level III biosafety laboratory (P3), which requires extremely high technical and safety measures, so it is impossible to achieve a large number of clinical samples and real-time detection. The antibody bLFA strip we established by selecting neutralizing mAb combined with colloidal gold, which can detect neutralizing antibodies in the serum samples and make a more accurate evaluation of the serum titer after clinical immunization, to obtain a more practical guiding significance.

## Figures and Tables

**Figure 1 vetsci-10-00152-f001:**
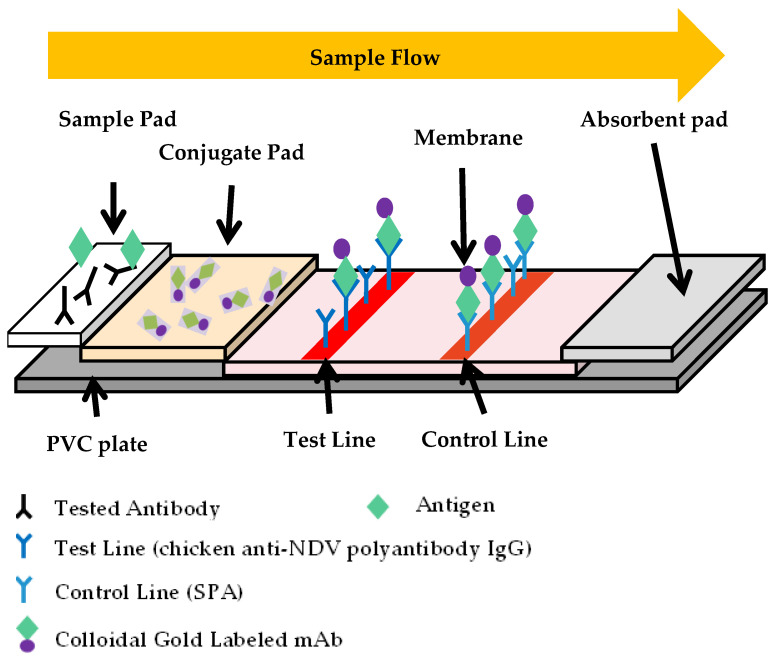
The schematic representation of the bLFA strip. The bLFA strip included three pads (sample, conjugate, and absorbent), a nitrocellulose (NC) membrane, and a PVC plate. The conjugate pad contained the dried gold-labeled mAb, which provided an easily visible red color. There were two blots of chicken anti-NDV polyantibody IgG and SPA for the T- and C-lines on the NC membrane, respectively.

**Figure 2 vetsci-10-00152-f002:**
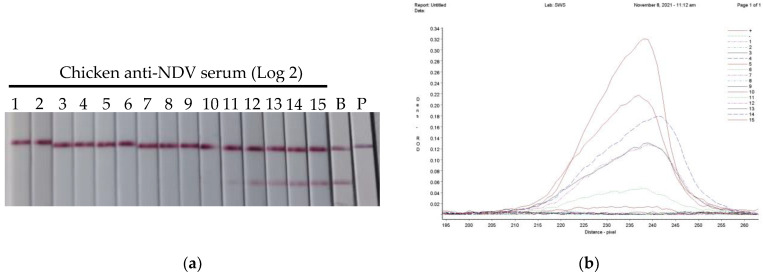
Determination of the bLFA strip sensitivity. The 2-fold serially diluted chicken NDV hyperimmunized serum sample was incubated with the HN protein solution for 10 min, and the mixtures were then applied to the blocking lateral flow bLFA strip and placed horizontally for 10 min to observe the result (**a**), and the colored membranes were screened by a TSR-3000 Reader, and the relative optical density (ROD) values of T-lines were analyzed by AIS software (**b**). Note: B: blank control is a strip containing protein and no serum samples; P: PBS control is a strip only containing phosphate-buffered saline.

**Figure 3 vetsci-10-00152-f003:**
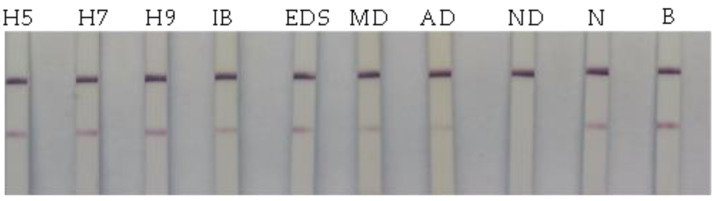
Determination of the bLFA strip specificity. The serum samples of the related avian viruses including AI (H5, H7, H9 subtypes), IB, EDS, MD, and AD, as well as ND positive serum and negative serum (N) were tested by the bLFA strip. Meanwhile, the HN detection antigen mixed with PBS is used as a blank control (B).

**Figure 4 vetsci-10-00152-f004:**
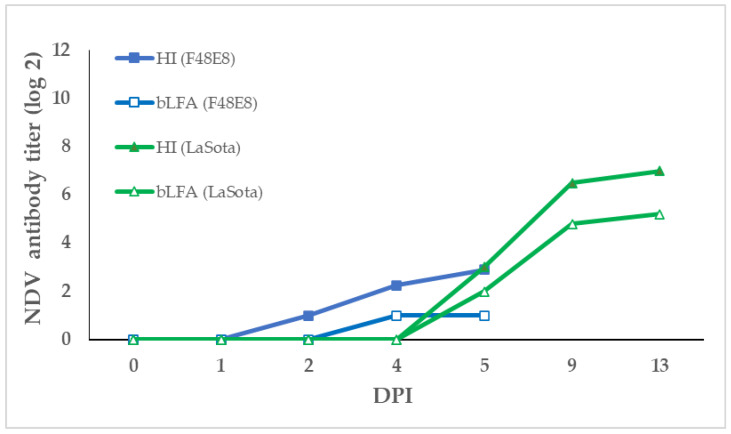
The dynamic change in NDV antibody in infected and vaccinated chickens. Serum samples were collected from the F_48_E_8_-infected and LaSota-vaccinated chickens at 1, 2, 4, 5, 9, and 13 DPI post-infected. The anti-NDV antibody titers of sera were then detected by the bLFA strip and HI test, respectively.

**Table 1 vetsci-10-00152-t001:** Screening of Monoclonal antibodies against HN protein for the bLFA.

mAbs	mAb Affinity for the Virus (LaSota) (Log 2) ^1^	bLFA for NDV Serum (Log 2) ^2^
1B3	0	/^3^
2A5	0	/
2C1	4	9
3G5	7	10
4D2	5	11
5D6	5	9
5E10	0	/
5F2	4	11
9C1	7	12
16G2	0	/

Notes: ^1^ virus titer of LaSota allantoic fluid was tested by the LFA strip using different gold-labeled mAbs; ^2^ antibody titer of NDV-specific serum was tested by the bLFA strip using different gold-labeled mAbs; ^3^ not test.

**Table 2 vetsci-10-00152-t002:** Optimized conditions of the bLFA strip for NDV antibody detection.

	Solution Concentration	Optimized Conditions
mAb conjugation	Ascitic fluid	50 µg/mL
Gold conjugate	10-time concentrated	5 µL/cm
Chicken anti-NDV polyantibody IgG (T-line)	1 mg/mL	5 µL/cm
SPA (C-line)	0.3 mg/mL	5 µL/cm
HN protein (dilution)	0.14 ng/µL	100 µL/well

**Table 3 vetsci-10-00152-t003:** Sensitivity of the bLFA strip compared with HI and VN tests.

Sample Dilution (Log 2)	bLFA Strip Test		HI Test	VN Test
	ROD	Result		
1	5.0558	+	+	+
2	5.5298	+	+	+
3	4.7175	+	+	+
4	5.4086	+	+	+
5	5.9615	+	+	+
6	4.2498	+	+	+
7	4.0453	+	+	+
8	4.6972	+	+	+
9	5.198	+	+	+
10	19.449	+	+	+
11	53.8691	+	+	+
12	133.7207	−	+	+
13	139.0825	−	+	+
14	189.8185	−	−	+
15	216.4233	−	−	−
Blank	278.3367	−	−	−
PBS	5.5128	−	−	−

Notes: + = positive; − = negative; HI = hemagglutination inhibition; PBS = phosphate-buffered saline: ROD = relative optical density; VN = virus neutralization; blank = containing HN protein without serum sample.

**Table 4 vetsci-10-00152-t004:** Sensitivity and specificity of the bLFA strip at different storage times.

Storage Time (mo)	Sensitivity	Specificity	
	Hyperimmunized Serum (HI 13 log 2)	AI H7 Positive Serum	Negative Serum
0	11 log 2	−^1^	−
3	11 log 2	−	−
6	11 log 2	−	−
9	11 log 2	−	−
12	11 log 2	−	−
18	10 log 2	−	−

Note: 1, negative result.

**Table 5 vetsci-10-00152-t005:** Clinical detection of anti-NDV antibody by the bLFA strip and HI test.

bLFA Strip	HI Test		Total
	Positive	Negative	
Positive	485 (TP)	0 (FP)	485
Negative	20 (FN)	13 (TN)	33
Total	505	13	510

Note: FN = false negative, FP = false positive, TN = true negative, TP = true positive.

## Data Availability

Not applicable.

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
