# Peer review of "Development and Evaluation of a Blocking Lateral Flow Assay Strip for Detection of Newcastle Disease Virus Antibodies"

_vetsci, 2023, doi:10.3390/vetsci10020152_

Round 1

Reviewer 1 Report

The authors describe a study about the development of a diagnostic for the detection of Newcastle Disease Virus (NDV) antibodies through a blocking lateral flow assay strip.

In general, the study is well described. With the climate change and globalization, it is expected a spread of NDV. Thus, the development of diagnostic applications for NDV will play an even more important role in the health of chickens. Therefore, I recommend this paper for publication with minor editing changes as suggested below:

Lines 136-137: “…measured according to the standard HA experiment procedure”

Which standard HA experiment procedure? Citation/Literature?

Line 171: “…3, 6, 9, 12, and 18 mo after…“

mo? Months? Please, reword this sentence.

Line 145: “…Briefly, 100 50% tissue culture infective dose…“

Which dose? 100 50%? Please, reword this sentence.

Line 192: “…Table 2 Optimized…” I´m missing a point between “2” and “Optimized”.

Line 225: Figure 2b is too small. It is not possible to read the AIS software analysis.

Line 232: “…Table 3 Sensitivity…” I´m missing a point between “3” and “Sensitivity”.

Line 265: Figure 4. I could not find figure 4 in the text.

Line 286: “…Table 5 clinical…” I´m missing a point between “5” and “Clinical”.

Lines 300-301: “…However, the routine HI and ELISA tests are not only complicated to operate, but also require 1-2 h detection time, so they are significantly better than HI and ELISA tests.”

Please, reword this sentence. Replace “they” with “bLFA strip” for better understanding.

Author Response

Dear Reviewer

We are thankful to your insightful comments. We have revised the manuscript in response to your suggestions.Please see the attachment.

Reviewer 2 Report

Manuscript comments

I thank you for considering me in the revision of this manuscript.

The work is innovative and addresses in a timely manner the need to generate a rapid technique for the detection of antibodies against Newcastle virus.

I believe that the publication can be accepted with some minimal recommendations.

Section 2. Material and methods

It is suggested that the following descriptions be moved from the Results section to Material and Methods.

Sample and antigen mixture preparation (amount of serum and amount and concentration of HN protein), described in 2020-2022 line.

Indicate that the specimen may contain antigen or antibody; however, the test only discriminates for antibody detection. In this topic, it is recommended to include in the discussion the recommended period in which the test can be performed after NDV infection (minimum antibody titers detected).

Describe in the sample and antigen mixture preparation step: amount of serum and amount and concentration of HN protein prior to sample placement on the sample pad, described in line 2020-2022.

Figure 1. It is recommended to include the drawings of antibodies and their interaction with the antigens that favor competitive binding in the conjugate area.

In the results section, line 198, it is indicated that the following techniques were performed: bLFA and LFA. In the material and methods section, the LFA procedure is not described, only the bLFA procedure is described.

Include in the material and methods the reader for quantitative measurement of antibodies. Specify the methodology to determine the limits of detection of antibodies by this method.

In the discussion section, include perspectives on the use of this new bLFIA method for antibody detection. Indicate when the test could be applied depending on the epidemiological situation of the farms in relation to the presence or absence of NDV infection. Also include the complementary tests that could be performed after the bLFIA that detects antibodies against NDV.

References section

References 5, 14 and 15 are exclusive publications in China. It is suggested to include more references that complement these citations, with publications of greater international availability.

Author Response

尊敬的审稿人

我们感谢您的深刻评论。我们根据您的建议修改了稿件。请参阅附件。
